# Retrospective Spatio-Temporal Dynamics of Dengue Virus 1, 2 and 4 in Paraguay

**DOI:** 10.3390/v15061275

**Published:** 2023-05-30

**Authors:** Cynthia Vazquez, Luiz Carlos Junior Alcantara, Vagner Fonseca, Mauricio Lima, Joilson Xavier, Talita Adelino, Hegger Fritsch, Emerson Castro, Carla de Oliveira, Gabriel Schuab, Alex Ranieri Jerônimo Lima, Shirley Villalba, Andrea Gomez de la Fuente, Analia Rojas, Cesar Cantero, Fatima Fleitas, Carolina Aquino, Andrea Ojeda, Guillermo Sequera, Juan Torales, Julio Barrios, Maria Carolina Elias, Felipe C. M. Iani, Maria Jose Ortega, Maria Liz Gamarra, Romeo Montoya, Evandra Strazza Rodrigues, Simone Kashima, Sandra Coccuzzo Sampaio, Norma Coluchi, Juliana Leite, Lionel Gresh, Leticia Franco, José Lourenço, Jairo Mendez Rico, Ana Maria Bispo de Filippis, Marta Giovanetti

**Affiliations:** 1Laboratorio Central de Salud Pública, Asunción 001535, Paraguay; cynthiavlm@yahoo.com (C.V.); shirleyvillalba@hotmail.com (S.V.); andre.gff585@gmail.com (A.G.d.l.F.); amnrojas@gmail.com (A.R.); cesarcantero24@gmail.com (C.C.); fatifleitas91@gmail.com (F.F.); lcala_py@hotmail.com (C.A.); juanbt15@hotmail.com (J.T.); biojulioc@gmail.com (J.B.); majosortega@yahoo.es (M.J.O.); malizga@yahoo.com (M.L.G.); ncoluchi@gmail.com (N.C.); 2Instituto Rene Rachou, Fundação Oswaldo Cruz, Belo Horizonte 30190-002, MG, Brazil; maurili15@hotmail.com (M.L.); joilsonxavier@live.com (J.X.); hegger.fritsch@gmail.com (H.F.); 3Organização Pan-Americana da Saúde, Organização Mundial da Saúde, Brasília 70-800400, SP, Brazil; vagner.fonseca@gmail.com; 4Laboratorio Central de Saúde Pública do Estado de Minas Gerais, Fundação Ezequiel Dias, Belo Horizonte 30510-010, MG, Brazil; talitaemile@yahoo.com.br (T.A.); emersoncb7@gmail.com (E.C.); felipeemrede@gmail.com (F.C.M.I.); 5Laboratório de Arbovírus e Vírus Hemorrágicos (LARBOH), Instituto Osawldo Cruz, Fiocruz, Rio de Janeiro 21040-900, RJ, Brazil; oliveirasc@yahoo.com.br (C.d.O.); gabrielshuab@gmail.com (G.S.); 6Butantan Institute, Center for Viral Surveillance and Serological Assessment (CeVIVAS), Sao Paulo 05507-000, SP, Brazil; alex.lima@butantan.gov.br (A.R.J.L.); carolina.eliassabbaga@butantan.gov.br (M.C.E.); sandra.coccuzzo@butantan.gov.br (S.C.S.); 7Dirección General de Vigilancia de la Salud, Asunción 40556, Paraguay; andy_bio2006@hotmail.com (A.O.); guillermo.sequera@mspbs.gov.py (G.S.); 8Enfermedades Trasmisibles y Determinantes Ambientales de la Salud CDE/HA/PHE, OPS/OMS, Asuncion 40126, Paraguay; montoyah@paho.org; 9Blood Center of Ribeirão Preto, Ribeirão Preto Medical School, University of São Paulo, Ribeirão Preto 14051-140, SP, Brazil; evandra@hemocentro.fmrp.usp.br (E.S.R.); skashima@hemocentro.fmrp.usp.br (S.K.); 10Emergency Department, Infectious Hazards Management, Pan American Health Organization, Washington, DC 20037, USA; leitejul@paho.org (J.L.); greshlio@paho.org (L.G.); francolet@paho.org (L.F.); ricoj@paho.org (J.M.R.); 11Biosystems and Integrative Sciences Institute, Faculty of Sciences, University of Lisbon, 1749-016 Lisbon, Portugal; jmlourenco@fc.ul.pt; 12Sciences and Technologies for Sustainable Development and One Health, University of Campus Bio-Medico, 00128 Rome, Italy

**Keywords:** Paraguay, DENV, genomic surveillance, phylodymamics

## Abstract

Dengue virus (DENV) has been a major public health concern in Paraguay, with frequent outbreaks occurring since early 1988. Although control measures have been implemented, dengue remains a significant health threat in the country, and continued efforts are required for prevention and control. In response to that, in collaboration with the Central Public Health Laboratory in Asunción, we conducted a portable whole-genome sequencing and phylodynamic analysis to investigate DENV viral strains circulating in Paraguay over the past epidemics. Our genomic surveillance activities revealed the co-circulation of multiple DENV serotypes: DENV-1 genotype V, the emerging DENV-2 genotype III, BR4-L2 clade, and DENV-4 genotype II. Results additionally highlight the possible role of Brazil as a source for the international dispersion of different viral strains to other countries in the Americas emphasizing the need for increased surveillance across the borders, for the early detection and response to outbreaks. This, in turn, emphasizes the critical role of genomic surveillance in monitoring and understanding arbovirus transmission and persistence locally and over long distances.

## 1. Introduction

Emerging and re-emerging arboviral diseases such as the dengue (DENV), Zika (ZIKV), and chikungunya (CHIKV) viruses pose a significant public health concern in Latin America, including Paraguay [1]. The country’s sub-tropical climate with high temperatures and humidity provides favorable conditions for the transmission of arboviruses, which are viruses transmitted to humans and other animals by blood-feeding arthropods such as mosquitoes [2]. Paraguay, a landlocked country in South America that shares borders with Brazil, Argentina, and Bolivia, has been severely affected by recent epidemics of ZIKV, CHIKV, and dengue fever [3].

DENV is the most common arboviral disease in Paraguay, with regular outbreaks occurring since 1988. All four DENV serotypes (DENV-1-4) have been detected in the country, with multiple serotypes co-circulating in some seasons. DENV-1 was first detected in 1988 and has been responsible for several epidemics in subsequent years [1]. DENV-2 was first isolated in Paraguay in 2001 and circulated at low transmission rates until 2005. Since 2010, the circulation of DENV-2 has been confirmed again and has caused significant epidemics over the years, resulting in an increase in severe cases and deaths [1]. DENV-3 was first detected during small dengue outbreaks in Paraguay in 2002 and 2003, and it has continued to circulate at a low level since then. As for DENV-4, it was first isolated in Paraguay in 2012, and it has continued to circulate seasonally without causing any major epidemics [1].

Despite the implementation of several measures to control the spread of arboviral diseases, little is still known regarding the viral landscape in the country [2]. Genomic surveillance is an extra layer towards control and outbreak response that provides key information about the diversity of circulating viral strains [4]. In this context, using a combination of portable whole-genome sequencing and genomic epidemiology, we found several cases of patients presenting febrile illness in different districts to be infected with different dengue virus serotypes, including DENV-1, DENV-2, and DENV-4. This study highlights the importance of active viral monitoring to prevent future epidemics, given the co-circulation of multiple DENV serotypes, and the crucial role of genomic surveillance in monitoring and understanding the transmission and persistence of arboviruses, both locally and over long distances.

## 2. Materials and Methods

Serum samples (*n* = 99) retrieved from patients presenting with symptoms compatible with an arboviral infection were collected by the Laboratorio Central de Salud Publica of Paraguay in Asunción for molecular diagnosis. Samples were submitted first to nucleic acid extraction using the QIAamp Viral RNA Mini Kit (QIAGEN) and then subjected to real-time reverse transcription PCR to detect ZIKV, CHIKV, and DENV serotypes 1–4 as described previously [5,6,7]. Positive samples were selected for sequencing based on the cycle threshold value ≤ 35 and the availability of demographic metadata such as sex, age, and municipality of residency. Extracted RNA was first converted to cDNA using the SuperScript IV Reverse Transcriptase kit (Invitrogen) and subjected to sequencing multiplex PCR (35 cycles) as previously described [8]. DNA library preparation was conducted using the Ligation Sequencing kit (Oxford Nanopore Technologies) and Native Barcoding Expansion 1–96 kit (Oxford Nanopore Technologies), following the reaction conditions as previously described in [8]. Sequencing was performed for up to 24 h on a MinION device and consensus sequences were obtained using the Genome Detective software [9].

The arbovirus genotyping tool was used to investigate sequence genotypes [10]. Subsequently, to analyze the origins and spatial dynamics of different dengue virus serotypes detected in Paraguay, we downloaded all sequences assigned as DENV-1, DENV-2, and DENV-4 from GenBank collected up to 15 March 2023. We excluded sequences without a sampling date and location as well as sequences that covered less than 50% of the virus genome.

All sequences were aligned using MAFFT [11] and manually edited to remove artifacts using Aliview [12]. The GTR nucleotide substitution model, which was inferred as the best-fit model by the ModelFinder application implemented in IQ-TREE2 [13], was used to estimate maximum Likelihood (ML) phylogenetic trees. The tree topology’s robustness was determined using 1000 bootstrap replicates. TempEst [14] was used to assess the presence of a temporal signal, and the BEAST package [15] was used to infer time-scaled phylogenetic trees. To estimate the most appropriate molecular clock model for the Bayesian phylogenetic analysis, we used a stringent model selection analysis that included both path-sampling (PS) and steppingstone (SS) procedures [16]. For all datasets, the uncorrelated relaxed molecular clock model was chosen by estimating marginal likelihoods using the codon-based SRD06 model of nucleotide substitution and the nonparametric Bayesian Skyline coalescent model. To model the phylogenetic diffusion of detected community transmission clades we used a flexible relaxed random walk diffusion model [17,18] that accommodates branch-specific variation in rates of dispersal with a Cauchy distribution and a jitter window site of 0.01 [19,20]. Latitude and longitude coordinates were assigned to each sequence. BEAST v1.10.4 was used for the MCMC analyses, which were run in duplicate for 100 million interactions and sampled every 10,000 steps in the chain. Tracer was used to assess convergence for each run (effective sample size for all relevant model parameters > 200). After discarding the initial 10% as burn-in, MCC trees for each run were summarized using TreeAnnotator. Finally, we extracted and mapped spatiotemporal information embedded in the posterior trees using the R package ‘seraphim’ version 1.0 [20].

Epidemiological time series data for confirmed, suspected, and probable DENV infections were downloaded from the PAHO website [21]. When presenting the series, we show “all cases” (all categories summed) and “confirmed” (confirmed category) separately. Index P estimations for Paraguay were obtained from Taishi et al. (2023) for Paraguay that were available per month and up to the year 2019 [22].

## 3. Results

To investigate the circulation of DENV-1, DENV-2, and DENV-4 in Paraguay, we generated a total of 99 DENV genome sequences (*n* = 47 DENV-1, *n* = 27 DENV-2, and *n* = 25 DENV-4) isolated from RT-qPCR positive samples using nanopore sequencing. With the exception of the Cordillera and Misiones regions, these samples were collected from 15 of Paraguay’s 17 departments between 2014 and 2022 (Figure 1A).

Seven of the seventeen departments reported the co-circulation of multiple viral strains (Figure 1A). The samples had a mean RT-qPCR cycle threshold value of 24, and their associated average age was 34 for females and 38 for males (years), with 57% of patients identified as female (Table 1). All patients were classified as having dengue with no warning signs, and no one was notified as severe.

Overall, the 99 novel DENV genome sequences had a mean genome coverage of 90.0% (coverage range 70–94%), with a mean depth of >1000× for all samples. All sequenced samples were collected between February 2014 and December 2022 and were made available on the GenBank database under the accession numbers OQ567774– OQ567872 (details in Table 1). The DENV typing tool classified all DENV-1 genomes generated here as genotype V, all DENV-2 genomes as genotype III, and all DENV-4 genomes as genotype II (Table 1).

Notified DENV cases revealed yearly epidemic waves recurrently taking place in the late summer months (Figure 1B). The estimated transmission potential, as measured by the mosquito-viral suitability index P, was found to capture the seasonal timing of notified cases over the years. However, these estimations were only available per month up to the year 2019. We compared the dynamics of three climatics (humidity, temperature, and precipitation) and index P versus notified DENV cases by calculating the Spearman’s correlation coefficient (Table 2). The highest coefficient was found between the index P and DENV notifications at 0.55, which increased to 0.71 when applying a lag of +1 month to the index P series (Figure 1B).

To explore the phylodynamic history of DENV-1, we then combined our 47 newly generated sequences with other DENV-1 genotype V genomes available on GenBank (*n* = 501). Phylogenetic analysis revealed that the novel isolates were organized into four distinct clades, named hereafter as clades I, II, III, and IV (Figure 2A).

Our analysis additionally suggested that the Brazilian southeast region was the leading hub responsible for multiple viral introductions in Paraguay. Among those clades, clade III resulted in a robust cluster, suggesting a likely persistence of that viral strain in the country over a period of 2 years (Figure 2A). To investigate the evolution of clade III in more detail, we used a smaller data set (*n* = 21) derived from this clade individually. An analysis of substitution rate constancy revealed a strong correlation between the sampling date and the root-to-tip divergence in this data set (r^2^ = 0.70, coefficient correlation= 0.84). Phylogeographic analyses of clade III allowed the reconstruction of viral movements across different districts in Paraguay (Figure 1B) and suggested a mean time of origin as mid-May 2015 (95% highest posterior density (HPD): 1 May 2015 to 6 June 2015). Viruses from this clade spread from the midwestern district (Distrito Capital) towards the southeast and later to the midwest, as indicated by isolates from the Presidente Hayes region (Figure 2A).

To explore the phylodynamic of DENV-2, we also performed a phylogenetic analysis of the 27 newly generated sequences plus 620 complete genome sequences of DENV-2 III available on GenBank (Figure 2B). Interestingly, our analysis allowed the detection of the emerging BR-4 L2 clade of DENV-2 genotype III in Paraguay. This clade was first detected at the end of 2019 in Brazil [21], and the introduction into Paraguay was likely mediated by the Brazilian Midwest region, reflecting a cross-border transmission that occurred in early July 2021 (95% highest posterior density (HPD): 21 December 2019 to 16 May 2021).

To explore the phylodynamic history of DENV-4, we performed a phylogenetic analysis of the 25 newly generated sequences plus 132 complete genome sequences of the DENV-4 genotype II available on GenBank (Figure 2C). We found that 96% (24 of 25) of the novel isolates from Paraguay fell within a single, large, well-supported monophyletic clade (bootstrap score BS = 100%). We also identified a single isolate outside the main clade. This isolate, sampled in March 2016, falls in a clade containing sequences from southeast Brazil, reinforcing the possible role of Brazil as a source for the international dispersion of different viral strains to other countries from the Americas. To gain more insight into the transmission dynamics of the large DENV-4 II clade from Paraguay, we built a dataset containing only isolates from Paraguay belonging to this clade (*n*= 33) and submitted it to phylodynamic inference. A regression of genetic divergence from root to tip against sampling dates confirmed that this subset had sufficient temporal signal (coefficient correlation = 0.82, r^2^ = 0.70). Phylogeographic analyses (Figure 2C) suggested a mean time of origin as late-April 2018 (95% highest posterior density (HPD): 16 April 2018 to 6 May 2018). Viruses from this clade spread from the midwestern district (Distrito Capital) towards the southeast and later to the northeast, as indicated by isolates from the Alto Parana region (Figure 2C).

## 4. Discussion

Since 1988, Paraguay has experienced significant dengue epidemics with multiple serotypes circulating concurrently during epidemic seasons. This increases the risk of secondary infections, which have a higher likelihood of severe illness and fatal outcomes [22].

DENV-1 was first detected in 1988 and has since been associated with ongoing epidemics [22]. DENV-2 was detected in 2001 and has caused significant epidemics linked to an increase in fatalities and severe cases since 2010 [22,23,24]. DENV-3 was first confirmed during small outbreaks in 2002 and 2003 and has since circulated at a low proportion, while DENV-4 was first reported in 2012 and has since circulated seasonally without causing significant epidemics [1]. The circulation of these serotypes has been characterized by multiple viral introduction events in the country, some of which were connected to viral strains circulating in other South American countries, such as Brazil [3].

Despite the co-circulation of different viral strains and the high disease burden, much remains unknown about the origins and routes of transmission of such viruses in the country. To obtain a better understanding of the DENV evolution in Paraguay, using whole genome sequencing and epidemiological analysis, we generated 99 nearly complete genome sequences collected between February 2014 and December 2022.

The newly generated DENV-1 sequences were classified as genotype V and formed four distinct clades (I-IV), one of which, clade III, resulted in a robust cluster, indicating the strain’s likely persistence in the country over a two-year period. Our findings also indicated that the Brazilian southeast region was the primary source of multiple viral introductions in Paraguay for this serotype. All new DENV-2 complete genome sequences obtained in this study were identified as belonging to genotype III, which has previously been found in South American epidemics. Interestingly, our analysis enabled the detection in Paraguay of the emerging DENV-2 genotype III, the BR-4 L2 clade, which was first detected in Brazil by the end of 2019. In addition, we found evidence of the co-circulation of the DENV-4 genotype II during the study period. Our findings indicated that most of the novel isolates from Paraguay from this serotype belonged to a single, large, well-supported monophyletic clade, implying the serotype’s persistence in the region over a three-year period. Surprisingly, this serotype is not the most common in other South American countries, including Brazil, where this viral strain has not been seen since 2020.

Notably DENV-3 was not detected in the study. However, it is unclear why DENV-3 is not currently circulating in Paraguay. One possibility is that the virus has been eradicated or reduced to such low levels that it is not detected by routine surveillance methods. Alternatively, DENV-3 may still be present in the country, but at a very low prevalence or in localized areas that were not sampled in the study. It is also possible that the epidemiological factors or transmission dynamics of DENV-3 in Paraguay are different from those of other serotypes, making it less likely to circulate or cause large outbreaks. Further research is necessary to gain a better understanding of the circulation and persistence of DENV-3 in Paraguay.

Overall, our study reveals a complex pattern of DENV transmission between epidemic seasons and sampling locations, with the southeastern and midwestern Brazilian regions acting as sources for international dispersion. Our findings highlight the need to increase genomic surveillance across borders to enable early detection and response to viral outbreaks. However, the scarcity of complete DENV genome sequences in South America limits our ability to characterize the molecular epidemiology of viral strains at a regional level, emphasizing the importance of increasing sequencing efforts to improve real-time data generation, sharing, and representativeness.

## Figures and Tables

**Figure 1 viruses-15-01275-f001:**
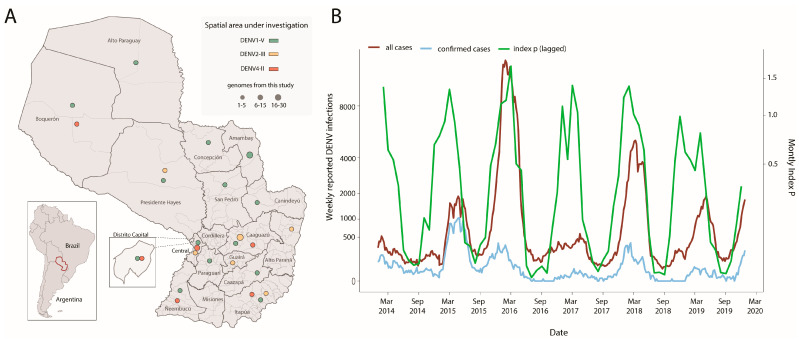
Spatio-temporal distribution of DENV-1, DENV-2, and DENV-4 in Paraguay. (**A**) Panel (**A**) displays a map of Paraguay with the sampling locations for dengue virus cases across all departments. The size of the circles represents the number of genomes isolated in each department, while the colors indicate the different viral strains detected in the country. The DENV-1 genotype V is highlighted in green; the DENV-2 genotype III in yellow, and the DENV-4 genotype II in orange; (**B**) Time series of weekly notified (red, suspected plus confirmed) and confirmed (light-blue) cases and monthly estimates of mosquito-viral suitability (index P) in Paraguay. Estimates of the index P were available on a monthly basis, but only up to the year 2019.

**Figure 2 viruses-15-01275-f002:**
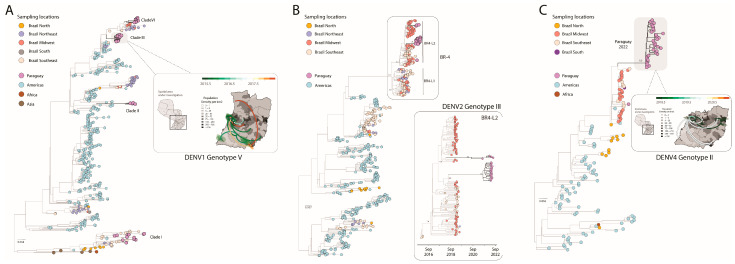
Dispersion dynamics of DENV-1, DENV-2, and DENV-4 in Paraguay. (**A**) Maximum likelihood (ML) phylogenetic analysis of 47 complete genome sequences from DENV-1 generated in this study plus other 501 available sequences from GenBank. The scale bar is in units of nucleotide substitutions per site (s/s) and the tree is mid-pointed rooted. Colors represent different sampling locations. The highlight on the right shows the phylogeographic reconstruction of Clade III (*n* = 21) in Paraguay. Circles represent nodes of the MCC phylogeny and are colored according to their inferred time of occurrence. Shaded areas represent the 80% highest posterior density interval and depict the uncertainty of the phylogeographic estimates for each node. Solid curved lines denote the links between nodes and the directionality of movement. Differences in population density are shown on a grey-white scale; (**B**) Maximum likelihood (ML) phylogenetic analysis of 27 complete genome sequences from DENV-2 generated in this study plus 620 available sequences from GenBank. The scale bar is in units of nucleotide substitutions per site (s/s) and the tree is mid-point rooted. Colors represent different sampling locations. The highlight on the right shows the MCC tree of the BR4 L2 clade. Values around key nodes represent posterior probability support; (**C**) Maximum likelihood (ML) phylogenetic analysis of 25 complete genome sequences from DENV4 genotype II generated in this study plus *n* = 132 available sequences from GenBank. The scale bar is in units of nucleotide substitutions per site (s/s) and the tree is mid-point rooted. Colors represent different sampling locations. The highlight on the right shows the phylogeographic reconstruction of the large DENV-4 II clade from Paraguay (*n* = 33). Circles represent nodes of the MCC phylogeny and are colored according to their inferred time of occurrence. Shaded areas represent the 80% highest posterior density interval and depict the uncertainty of the phylogeographic estimates for each node. Solid curved lines denote the links between nodes and the directionality of movement. Differences in population density are shown on a grey-white scale.

**Table 1 viruses-15-01275-t001:** Epidemiological data of the 99 DENV1-2 and 4 genome sequences obtained as part of this study.

Accession Number	Virus	Lineages	Country	Ct	City	Collection Date	Date Onset Symptoms	Gender	Age	Coverage
OQ567774	DENV-1	Genotype V	PY	24	Salto Del Guaira	27-Feb-2014	27-Feb-2014	M	27	88.7
OQ567775	DENV-1	Genotype V	PY	22	Santa Rosa Del Aguaray	23-Mar-2015	20-Mar-2015	M	28	88.5
OQ567776	DENV-1	Genotype V	PY	20	Santa Rosa Del Aguaray	20-Mar-2015	18-Mar-2015	F	41	93.6
OQ567777	DENV-1	Genotype V	PY	20	Santa Rosa Del Aguaray	26-Mar-2015	25-Mar-2015	F	27	93.2
OQ567778	DENV-1	Genotype V	PY	24	Carapegua	13-May-2015	12-May-2015	M	43	89.3
OQ567779	DENV-1	Genotype V	PY	26	Caazapa	2-Jun-2015	31-May-2015	M	46	89.2
OQ567780	DENV-1	Genotype V	PY	18	Ciudad del Este	15-Dec-2015	14-Dec-2015	M	41	93
OQ567781	DENV-1	Genotype V	PY	19	Asuncion	18-Dec-2015	17-Dec-2015	F	22	93.4
OQ567782	DENV-1	Genotype V	PY	20	Concepcion	17-Dec-2015	14-Dec-2015	F	36	94.4
OQ567783	DENV-1	Genotype V	PY	22	Concepcion	23-Dec-2015	21-Dec-2015	F	53	88.8
OQ567784	DENV-1	Genotype V	PY	21	Benjamin Aceval	21-Dec-2015	20-Dec-2015	M	SD	88.8
OQ567785	DENV-1	Genotype V	PY	25	San Juan Nepomuceno	24-Dec-2015	22-Dec-2015	F	19	88.7
OQ567786	DENV-1	Genotype V	PY	27	Itapua	29-Jan-2016	26-Jan-2016	M	18	88.5
OQ567787	DENV-1	Genotype V	PY	22	Pilar	4-Feb-2016	2-Feb-2016	F	24	88.8
OQ567788	DENV-1	Genotype V	PY	25	Pilar	2-Apr-2016	2-Mar-2016	F	24	92.9
OQ567789	DENV-1	Genotype V	PY	26	Yatytay	2-Jan-2016	2-Jan-2016	F	83	92.9
OQ567790	DENV-1	Genotype V	PY	26	San Lorenzo	24-Jan-2016	23-Jan-2016	F	23	92.6
OQ567791	DENV-1	Genotype V	PY	28	Asuncion	10-Feb-2016	2-Jun-2016	F	28	89
OQ567792	DENV-1	Genotype V	PY	26	Carapegua	2-Sep-2016	2-Aug-2016	M	18	92.7
OQ567793	DENV-1	Genotype V	PY	21	Pirayu	2-Sep-2016	2-Jun-2016	F	40	94.1
OQ567794	DENV-1	Genotype V	PY	28	Coronel Oviedo	2-Oct-2016	2-Aug-2016	F	37	84.1
OQ567795	DENV-1	Genotype V	PY	22	Itaugua	14-Feb-2016	14-Feb-2016	F	23	93.1
OQ567796	DENV-1	Genotype V	PY	26	Mariscal Estigarribia	3-Jan-2017	28-Feb-2017	F	61	92.9
OQ567797	DENV-1	Genotype V	PY	27	Asuncion	29-Mar-2017	28-Mar-2017	F	29	89.2
OQ567798	DENV-1	Genotype V	PY	21	Paraguari	26-Apr-2017	25-Apr-2017	F	28	89.3
OQ567799	DENV-1	Genotype V	PY	21	Tte. Irala Fernandez	25-Apr-2017	23-Apr-2017	F	44	93.3
OQ567800	DENV-1	Genotype V	PY	24	Filadeflia	27-Mar-2017	26-Mar-2017	M	25	92.9
OQ567801	DENV-1	Genotype V	PY	27	San Lorenzo	29-Apr-2017	26-Apr-2017	F	25	88.9
OQ567802	DENV-1	Genotype V	PY	27	Capitan Bado	22-Mar-2022	21-Mar-2022	F	18	93
OQ567803	DENV-1	Genotype V	PY	22	Capitan Bado	28-Mar-2022	26-Mar-2022	M	18	89.6
OQ567804	DENV-1	Genotype V	PY	25	Capitan Bado	4-Apr-2022	1-Apr-2022	M	56	88.1
OQ567805	DENV-1	Genotype V	PY	26	Guarambare	25-Mar-2022	23-Mar-2022	M	41	94.6
OQ567806	DENV-1	Genotype V	PY	26	Capitan Bado	4-Dec-2022	11-Apr-2022	F	57	91.8
OQ567807	DENV-1	Genotype V	PY	28	Capitan Bado	19-Apr-2022	15-Apr-2022	M	52	71
OQ567808	DENV-1	Genotype V	PY	26	Capitan Bado	18-Apr-2022	16-Apr-2022	M	20	75.1
OQ567809	DENV-1	Genotype V	PY	21	Capitan Bado	21-Apr-2022	19-Apr-2022	F	11	83.9
OQ567810	DENV-1	Genotype V	PY	28	Capitan Bado	22-Apr-2022	20-Apr-2022	M	36	88.8
OQ567811	DENV-1	Genotype V	PY	22	Capitan Bado	21-Apr-2022	17-Apr-2022	M	61	93.8
OQ567812	DENV-1	Genotype V	PY	26	Capitan Bado	20-Apr-2022	16-Apr-2022	M	33	94.2
OQ567813	DENV-1	Genotype V	PY	27	Capitan Bado	20-Apr-2022	17-Apr-2022	M	61	93.4
OQ567814	DENV-1	Genotype V	PY	21	Encarnacion	4-May-2022	2-May-2022	F	18	94.2
OQ567815	DENV-1	Genotype V	PY	21	Villa Ygatimi	10-May-2022	7-May-2022	F	62	88.3
OQ567816	DENV-1	Genotype V	PY	24	Capitan Bado	4-May-2022	2-May-2022	F	18	79.4
OQ567817	DENV-1	Genotype V	PY	27	Pedro Juan Caballero	5-Jun-2022	1-Jun-2022	F	37	93.5
OQ567818	DENV-1	Genotype V	PY	27	Salto del Guaira	2-Jun-2022	1-Jun-2022	F	14	94.2
OQ567819	DENV-1	Genotype V	PY	22	Salto del Guaira	20-Jun-2022	16-Jun-2022	M	72	93
OQ567820	DENV-1	Genotype V	PY	25	Capitan Bado	14-Jun-2022	11-Jun-2022	F	17	84.7
OQ567821	DENV-2	Genotype III	PY	26	Villarrica	2-Jul-2021	1-Jul-2021	M	29	75.3
OQ567822	DENV-2	Genotype III	PY	26	Mariano Roque Alonso	10-Jul-2021	10-Jul-2021	F	21	70
OQ567823	DENV-2	Genotype III	PY	28	Coronel Oviedo	28-Mar-2022	26-Mar-2022	M	35	90.9
OQ567824	DENV-2	Genotype III	PY	26	Coronel Oviedo	27-Mar-2022	24-Mar-2022	M	49	93.1
OQ567825	DENV-2	Genotype III	PY	21	Coronel Oviedo	29-Mar-2022	27-Mar-2022	M	23	91.1
OQ567826	DENV-2	Genotype III	PY	28	Coronel Oviedo	5-Apr-2022	2-Apr-2022	M	28	93.1
OQ567827	DENV-2	Genotype III	PY	22	Coronel Oviedo	11-Apr-2022	9-Apr-2022	F	33	87.2
OQ567828	DENV-2	Genotype III	PY	26	Coronel Oviedo	7-Apr-2022	5-Apr-2022	F	29	78.6
OQ567829	DENV-2	Genotype III	PY	27	Coronel Oviedo	17-Apr-2022	14-Apr-2022	M	22	93
OQ567830	DENV-2	Genotype III	PY	21	Coronel Oviedo	18-Apr-2022	16-Apr-2022	F	48	93.2
OQ567831	DENV-2	Genotype III	PY	21	Caaguazu	22-Apr-2022	20-Apr-2022	M	61	93.6
OQ567832	DENV-2	Genotype III	PY	24	Coronel Oviedo	28-Apr-2022	26-Apr-2022	F	45	71.1
OQ567833	DENV-2	Genotype III	PY	27	Coronel Oviedo	2-May-2022	1-May-2022	M	51	93.3
OQ567834	DENV-2	Genotype III	PY	27	Coronel Oviedo	27-Apr-2022	25-Apr-2022	M	59	84.4
OQ567835	DENV-2	Genotype III	PY	22	Coronel Oviedo	6-May-2022	3-May-2022	F	42	90.7
OQ567836	DENV-2	Genotype III	PY	25	Coronel Oviedo	9-May-2022	5-May-2022	M	51	89.8
OQ567837	DENV-2	Genotype III	PY	26	Coronel Oviedo	8-May-2022	6-May-2022	F	43	86.8
OQ567838	DENV-2	Genotype III	PY	26	Coronel Oviedo	22-May-2022	18-May-2022	F	37	89.8
OQ567839	DENV-2	Genotype III	PY	28	Coronel Oviedo	29-May-2022	19-May-2022	F	26	92
OQ567840	DENV-2	Genotype III	PY	26	Repatriacion	24-Jun-2022	23-Jun-2022	F	44	88.4
OQ567841	DENV-2	Genotype III	PY	21	Benjamin Aceval	23-Feb-2022	22-Feb-2022	M	14	78.8
OQ567842	DENV-2	Genotype III	PY	28	Coronel Oviedo	22-Mar-2022	20-Mar-2022	F	46	92.9
OQ567843	DENV-2	Genotype III	PY	22	Coronel Oviedo	5-May-2022	4-May-2022	M	12	70.2
OQ567844	DENV-2	Genotype III	PY	26	Coronel Oviedo	4-May-2022	4-May-2022	M	24	93.2
OQ567845	DENV-2	Genotype III	PY	27	Capiata	27-Apr-2022	24-Apr-2022	F	20	87.2
OQ567846	DENV-2	Genotype III	PY	21	Presidente Franco	24-Mar-2021	22-Mar-2021	F	61	92.8
OQ567847	DENV-2	Genotype III	PY	21	Encarnacion	14-Apr-2021	4-Dec-2021	M	59	92.7
OQ567848	DENV-4	Genotype II	PY	24	Coronel Oviedo	7-Mar-2016	7-Mar-2016	F	37	91.9
OQ567849	DENV-4	Genotype II	PY	27	Asuncion	27-Jan-2020	20-Jan-2024	F	47	92.2
OQ567850	DENV-4	Genotype II	PY	24	Asuncion	30-Jan-2020	28-Jan-2020	F	14	92
OQ567851	DENV-4	Genotype II	PY	28	Filadelfia	1-Feb-2020	1-Feb-2020	M	11	92
OQ567852	DENV-4	Genotype II	PY	20	Lambare	4-Feb-2020	2-Apr-2020	M	29	92.5
OQ567853	DENV-4	Genotype II	PY	27	Asuncion	6-Feb-2020	4-Feb-2020	M	30	91.7
OQ567854	DENV-4	Genotype II	PY	23	Pilar	4-Feb-2020	4-Feb-2020	F	60	92
OQ567855	DENV-4	Genotype II	PY	26	Itaugua	7-Feb-2020	3-Feb-2020	F	40	92.5
OQ567856	DENV-4	Genotype II	PY	26	Itagua	10-Feb-2020	8-Feb-2020	F	77	92.5
OQ567857	DENV-4	Genotype II	PY	29	Obligado	13-Feb-2020	2-Nov-2020	F	4	92
OQ567858	DENV-4	Genotype II	PY	26	Itagua	1-Jan-2020	31-Dec-2019	F	29	92
OQ567859	DENV-4	Genotype II	PY	21	Lambare	7-Jan-2020	1-May-2020	M	31	93
OQ567860	DENV-4	Genotype II	PY	25	Villa Elisa	7-Jan-2020	1-Apr-2020	F	32	92.6
OQ567861	DENV-4	Genotype II	PY	17	Nemby	9-Jan-2020	1-Sep-2020	M	1M	92.5
OQ567862	DENV-4	Genotype II	PY	27	Itagua	12-Jan-2020	1-Nov-2020	F	30	91.8
OQ567863	DENV-4	Genotype II	PY	30	Nemby	13-Jan-2020	1-Dec-2020	F	29	91.9
OQ567864	DENV-4	Genotype II	PY	21	Trinidad	13-Jan-2020	13-Jan-2020	M	47	92.4
OQ567865	DENV-4	Genotype II	PY	25	Coronel Oviedo	11-Jan-2020	1-Nov-2020	M	59	92.6
OQ567866	DENV-4	Genotype II	PY	30	San Lorenzo	12-Jan-2020	1-Dec-2020	F	77	92
OQ567867	DENV-4	Genotype II	PY	23	San Lorenzo	17-Jan-2020	15-Jan-2020	F	37	92.4
OQ567868	DENV-4	Genotype II	PY	23	Ypane	18-Jan-2020	12-Jan-2020	M	25	92.1
OQ567869	DENV-4	Genotype II	PY	19	San Lorenzo	15-Jan-2020	11-Jan-2020	M	5M	92.7
OQ567870	DENV-4	Genotype II	PY	26	Itagua	17-Jan-2020	12-Jan-2020	F	18	92.9
OQ567871	DENV-4	Genotype II	PY	20	San Lorenzo	20-Jan-2020	15-Jan-2020	F	1	92.5
OQ567872	DENV-4	Genotype II	PY	30	Pilar	22-Feb-2021	19-Feb-2021	F	7	92.1

**Table 2 viruses-15-01275-t002:** Monthly variable correlations with the number of notified DENV cas.es.

Variable 1	Variable 2	Correlation (Spearman’s Coefficient)
index p	total cases *	0.55
temperature	total cases *	0.37
humidity	total cases *	0.50
precipitation	total cases *	0.40
index p	confirmed cases	0.51
temperature	confirmed cases	0.34
humidity	confirmed cases	0.43
precipitation	confirmed cases	0.42
index p (lagged)	total cases *	0.71
index p (lagged)	confirmed cases	0.62

* Total cases = suspected plus confirmed. Cases were log10 transformed before calculating the Spearman’s coefficient.

## Data Availability

Newly generated sequences have been deposited in GenBank under accession numbers OQ567774–OQ567872.

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
