# Peer review of "Retrospective Spatio-Temporal Dynamics of Dengue Virus 1, 2 and 4 in Paraguay"

_viruses, 2023, doi:10.3390/v15061275_

Round 1
Reviewer 1 Report
Vazquez et al. present results regarding phylogenetic analysis of Dengue Virus in Paraguay, which is quite important, such Latin American countries lacks information of patterns of spread for arboviruses. However, it was quite difficult to analyse properly the manuscript since the figures are in a low quality. I recommend authors to present a new version of the manuscript with figures in a better definition.
Author Response
Dear Editor,
We are pleased that the editorial assessment of our manuscript was largely favourable and thank the reviewers and editors for the constructive comments. We have addressed all comments (see below) and we believe that the resulting manuscript is much improved. We hope you will consider the revision to be suitable for publication in Viruses.
Reviewer 1:
Comment 1: Vazquez et al. present results regarding phylogenetic analysis of Dengue Virus in Paraguay, which is quite important, such Latin American countries lacks information of patterns of spread for arboviruses. However, it was quite difficult to analyse properly the manuscript since the figures are in a low quality. I recommend authors to present a new version of the manuscript with figures in a better definition.
Reply: We thank the reviewer for the positive comment. We appreciate your valuable feedback and have followed your suggestion by uploading our figures into the online system as high-resolution .tiff files.
Reviewer 2 Report
Viruses-2372110-peer-review-comments-v1
Brief Report
Retrospective spatio-temporal dynamics of Dengue virus 1, 2 and 4 in Paraguay
Cynthia Vazquez et.al.
Overall Comments :
The paper presents a study on the genomic surveillance of the Dengue virus (DENV) in Paraguay. Considering the fact that dengue is a public health problem of major concern in Paraguay, and frequent outbreaks have been occurring since early 1988, the study is important for the country for epidemiological molecular surveillance to understand the arbovirus transmission and its persistence in different areas of the country. It is not clear from the study why DENV 3 was not found in Paraguay. The authors have not mentioned in the introduction as well as discussion anything regarding the presence or absence of this serotype in previously published studies as well as the present study. It will be desirable to add information on the same. Since the study is expected to improve the public health interventions for the prevention and control of Dengue in Paraguay, the discussion should lead to guidance on such matters.
I find the paper quite informative and recommend it for publication after incorporating the following minor corrections, clarifications and changes.
Abstract :
Keywords : Please consider replacing ‘Phylodynamic’ with ‘Phylodymamics’.
Introduction : Please add briefly the available information on DENV 3 in Paraguay.
Material and methods:
Though most of the important information has been covered under results, it will be good to add briefly the criteria based on which 99 samples were selected for genotyping. Would the authors be able to say that these 99 samples represented an epidemiologically significant sample size for representing the study areas?
Results:
· Please clarify in the paper for the benefit of the readers what is meant by ‘departments’ from where samples were collected.
· Check the sentence ‘With the exception of the Cordillera and Misiones regions, these genomes were collected from 15 of Paraguay's 17 departments between 2014 and 2022 (Figure 1A)’. Is it genomes or samples? Also, please check the difference in the years in Figure 1B. The years represented in the figure are from 2014 to 2020 instead of 2022.
Discussion :
The DENV 3 has not been reported in the study. Please add a discussion on the reasons for the absence of this strain in the study. If it has never been reported from Paraguay please mention accordingly and discuss the possible reasons also. Please also mention here the prevailing guidelines for dengue genotyping in Paraguay and how systematic genotyping will be useful for better prevention/control of dengue in the country and for strengthening public health actions. This will add value to the study from a public health point of view.
References :
References are written in different formats and need revision. Please follow a uniform style of writing the reference as per the recommendations of the journal.

Author Response
Dear Editor,
We are pleased that the editorial assessment of our manuscript was largely favourable and thank the reviewers and editors for the constructive comments. We have addressed all comments (see below) and we believe that the resulting manuscript is much improved. We hope you will consider the revision to be suitable for publication in Viruses.
Reviewer 2:
Overall Comments: The paper presents a study on the genomic surveillance of the Dengue virus (DENV) in Paraguay. Considering the fact that dengue is a public health problem of major concern in Paraguay, and frequent outbreaks have been occurring since early 1988, the study is important for the country for epidemiological molecular surveillance to understand the arbovirus transmission and its persistence in different areas of the country. It is not clear from the study why DENV 3 was not found in Paraguay. The authors have not mentioned in the introduction as well as discussion anything regarding the presence or absence of this serotype in previously published studies as well as the present study. It will be desirable to add information on the same. Since the study is expected to improve the public health interventions for the prevention and control of Dengue in Paraguay, the discussion should lead to guidance on such matters. I find the paper quite informative and recommend it for publication after incorporating the following minor corrections, clarifications, and changes.
Reply: We appreciate the reviewer's positive comment. We agree that the paper lacked information on DENV-3 circulation in Paraguay. Therefore, we have updated the introduction section to provide additional information on the circulation of DENV-3 in Paraguay.
Comment 1: Keywords: Please consider replacing ‘Phylodynamic’ with ‘Phylodymamics’.
Reply: Done.
Comment 2: Introduction: Please briefly add the available information on DENV 3 in Paraguay.
Reply: Done.
Comment 3: Material and methods: Though most of the important information has been covered under results, it will be good to add briefly the criteria based on which 99 samples were selected for genotyping. Would the authors be able to say that these 99 samples represented an epidemiologically significant sample size for representing the study areas?
Reply: We appreciate the reviewer comment and we have now made some changes to this section which now reads:
“Serum samples (n=99) retrieved from patients presenting symptoms compatible with arboviral infection were collected by the Laboratorio Central de Salud Publica of Paraguay, in Asunción for molecular diagnosis. Samples were submitted first to nucleic acid extraction using the QIAamp Viral RNA Mini Kit (QIAGEN) and then subjected to real-time reverse transcription PCR to detect ZIKV, CHIKV and DENV serotypes 1–4 as described previously [5-7]. Positive samples were selected for sequencing based on the cycle threshold value ≤ 35 and the availability of demographic metadata such as sex, age, and municipality of residency.”
Results:
Comment 4: Please clarify in the paper for the benefit of the readers what is meant by ‘departments’ from where samples were collected.
Reply: We appreciated the reviewer comment, and we are keen to apply any change to the text that may reduce misinterpretations. However, we would like to clarify that this term is commonly used by the Paraguayan Institute of Geography and Statistics to describe the 17 administrative divisions of Paraguay.
Comment 5: Check the sentence ‘With the exception of the Cordillera and Misiones regions, these genomes were collected from 15 of Paraguay's 17 departments between 2014 and 2022 (Figure 1A)’. Is it genomes or samples? Also, please check the difference in the years in Figure 1B. The years represented in the figure are from 2014 to 2020 instead of 2022.
Reply: Thank you to the reviewer for bringing this to our attention. We have updated the text to replace "genomes" with "samples." Regarding figure 1B, as stated in the methods section, unfortunately, Index P estimations for Paraguay were available up to the year 2019.
Comment 6: Discussion: The DENV 3 has not been reported in the study. Please add a discussion on the reasons for the absence of this strain in the study. If it has never been reported from Paraguay please mention accordingly and discuss the possible reasons also. Please also mention here the prevailing guidelines for dengue genotyping in Paraguay and how systematic genotyping will be useful for better prevention/control of dengue in the country and for strengthening public health actions. This will add value to the study from a public health point of view.
Reply: We thank the reviewer for this comment. According to the epidemiological data, DENV-3 was first detected in Paraguay during small dengue outbreaks in 2002 and 2003. Since then, this serotype has been circulating at a low level without causing major epidemics. However, it is unclear why DENV-3 is not currently circulating in Paraguay. One possibility is that the virus has been eradicated or reduced to such low levels that it is not detected by routine surveillance methods. Alternatively, DENV-3 may still be present in the country, but at a very low prevalence or in localized areas that were not sampled in the study. It is also possible that the epidemiological factors or transmission dynamics of DENV-3 in Paraguay are different from those of other serotypes, making it less likely to circulate or cause large outbreaks. Further research is necessary to gain a better understanding of the circulation and persistence of DENV-3 in Paraguay. We have now provided additional information in the text in the introduction and discussion sections.
Comment 7: References: References are written in different formats and need revision. Please follow a uniform style of writing the reference as per the recommendations of the journal.
Reply: Done.